# Pharmacotherapeutic Approaches to Treatment of Muscular Dystrophies

**DOI:** 10.3390/biom13101536

**Published:** 2023-10-17

**Authors:** Alan Rawls, Bridget K. Diviak, Cameron I. Smith, Grant W. Severson, Sofia A. Acosta, Jeanne Wilson-Rawls

**Affiliations:** 1School of Life Sciences, Arizona State University, Tempe, AZ 85287-4501, USA; bdiviak@asu.edu (B.K.D.); cismith@asu.edu (C.I.S.); gwsevers@asu.edu (G.W.S.); saacost6@asu.edu (S.A.A.); 2Molecular and Cellular Biology Graduate Program, School of Life Sciences, Tempe, AZ 85287 4501, USA

**Keywords:** muscular dystrophy, dystrophin, dysferlin, DUX4, dystroglycan, inflammation, fibrosis, sarcoglycan, lamin A, emerin

## Abstract

Muscular dystrophies are a heterogeneous group of genetic muscle-wasting disorders that are subdivided based on the region of the body impacted by muscle weakness as well as the functional activity of the underlying genetic mutations. A common feature of the pathophysiology of muscular dystrophies is chronic inflammation associated with the replacement of muscle mass with fibrotic scarring. With the progression of these disorders, many patients suffer cardiomyopathies with fibrosis of the cardiac tissue. Anti-inflammatory glucocorticoids represent the standard of care for Duchenne muscular dystrophy, the most common muscular dystrophy worldwide; however, long-term exposure to glucocorticoids results in highly adverse side effects, limiting their use. Thus, it is important to develop new pharmacotherapeutic approaches to limit inflammation and fibrosis to reduce muscle damage and promote repair. Here, we examine the pathophysiology, genetic background, and emerging therapeutic strategies for muscular dystrophies.

## 1. Introduction

Muscular dystrophies (MDs) are genetic degenerative neuromuscular diseases characterized by progressive muscle weakness that results in significant morbidity. To date, mutations in 57 genes have been identified that cause nine specific classes of muscular dystrophy [1]. Dystrophies are classified based on the specific gene involved and clinical features such as muscles affected, rate of disease progression, histopathology, and age of diagnosis. Inflammation is a common factor in the pathogenesis and progression of many types of MD. Chronic inflammation exacerbates muscle damage, induces fibrotic deposition and fatty replacement of myofibers, and impedes the regenerative process of skeletal muscle.

There are no curative treatments for any dystrophies currently, but new AAV and base editing approaches to provide the missing proteins or fix the genetic lesion provide hope. Though there is tremendous curative potential, these genetic approaches will not treat all dystrophies or even all versions of a single dystrophy. For example, there are more than 7000 known mutations in dystrophin that result in Duchenne muscular dystrophy (DMD) [2]. Therefore, understanding the mechanisms underlying inflammation and other pathogenic processes will aid in identifying therapeutic approaches that can ameliorate the progression of these diseases. Preclinical and clinical treatments that specifically target inflammation and fibrosis are the subject of this review.

## 2. Duchenne and Becker Muscular Dystrophy

Duchenne muscular dystrophy (DMD), the most common and severe form of MD, is caused by mutations in the dystrophin (*DMD*) gene (Figure 1). This gene is located on the X chromosome [3], and DMD affects 1 in 3600 males born [4]. Clinically, DMD is characterized by muscle weakness and wasting during early childhood, with symptoms appearing by age 3. Initially, these patients have difficulty or pain with movement, frequent falls, a waddling gait, and delayed growth [5]. There is a loss of ambulation in childhood to early teens and death in early adulthood [6,7]. DMD patients also suffer significant cardiac complications; approximately 90% will progress to dilated cardiomyopathy marked by inflammatory cell infiltration, fibrosis, and myocardial cell death, leading to early mortality [8,9]. In contrast, Becker MD (BMD), the second most common form of muscular dystrophy, also caused by mutations in *DMD*, is found in 1.5–3.6/100,000 males globally. BMD is a milder disease, and patients typically are identified at 11–25 years of age; however, they may have much later ages of onset. BMD patients develop progressive weakness in the muscles of the hips, thighs, pelvis, and shoulders and generally have a nearly normal lifespan unless they experience cardiac failure [10].

*DMD* is the largest gene in the human genome; it has 79 exons, and the full-length transcripts include unique tissue-specific first exons [11,12]. The dystrophin protein is 427 kDa in size, it consists of four domains; at the N-terminus, there is an actin-binding domain, which is followed by a central rod domain consisting of 24 spectrin-like repeats. Within the rod domain are four hinge regions at the N- and C- termini of the rod. The fourth hinge region is followed by the cysteine-rich domain that contains a dystroglycan binding site. At the C-terminus of dystrophin, there are α-syntrophin and dystrobrevin binding sites [13,14]. Dystrophin is part of a large multiprotein complex that acts to stabilize the myofiber membrane during muscle contraction to prevent contraction-induced damage [13,14]. The dystrophin–glycoprotein complex (DGC) includes the cytoplasmic proteins dystrobrevin and syntrophins, sarcolemmal transmembrane proteins β-dystroglycan, sarcoglycans, sarcospan, and an extracellular protein α-dystroglycan [14]. Dystrophin binds directly to dystroglycan, dystrobrevin, and α-syntrophin at its C-terminus, while the N-terminal region binds actin intracellularly. These connections link the contractile apparatus through actin to the membrane and extracellular matrix [13,15,16], allowing the DGC to mediate mechanical force signal transduction and cell adhesion-based signaling [17].

Loss of membrane integrity and stability in skeletal and cardiac muscle due to a lack of dystrophin underlies DMD pathology (Figure 1) [13,14,18]. There are thousands of mutations that are known to cause DMD/BMD. In DMD, 60–70% of the mutations are large deletions that span one or more exons, whereas approximately 26% are point mutations that result in frameshifts or premature termination [14]. The milder disease course and later onset of BMD are the result of mutations that maintain the reading frame while deleting exons in the midportion of the rod domain, resulting in a shorter but functional dystrophin protein [10,18].

Inflammation is necessary for the normal skeletal muscle repair process, but muscle damage in DMD/BMD results in a chronic inflammatory environment. In skeletal and cardiac muscle, functional muscle loss, including diaphragmatic weakness and respiratory insufficiency, is linked to fibrosis that is caused by chronic inflammation [8,9,19]. As muscle wasting progressively increases over time, skeletal muscle is replaced by extracellular matrix (ECM), resulting in fibrosis that further hinders the ability of dystrophic muscle to generate the necessary force. A key factor controlling fibrosis is transforming growth factor β (TGF-β), which is stored in the ECM. It promotes the synthesis and remodeling of ECM proteins during muscle regeneration. In dystrophic muscle, TGF-β is activated chronically and induces the deposition of fibrotic proteins such as collagen to replace degenerated muscle. Reducing fibrosis has been explored as a treatment avenue for DMD. Potential targets include TGF-β and connective tissue growth factor (CTGF) [20]. Immunosuppression with glucocorticoids, such as deflazacort or prednisone, is the current guideline treatment for DMD symptoms. Glucocorticoids are effective in reducing symptoms and slowing disease progression but are associated with many adverse side effects, including uneven and excessive weight gain, loss of bone density, reduced stature, behavioral changes, and even muscular atrophy, which all negatively impact the quality of life for patients [21,22,23]. New strategies that inhibit inflammation and fibrosis and improve muscle function are essential. Below, we discuss some potential therapeutics, grouped by strategy, that have shown promise but are not FDA approved currently (summarized in Table 1).

### 2.1. Inhibition of Inflammation and Fibrosis

Edasalonexent (CAT-1004, is a small molecule NF-κB inhibitor that combines structural elements of salicylic acid and docosahexenoic acid [24]. A phase 3 placebo-controlled clinical trial of this drug in DMD boys (aged 4–8 years) found it to be safe and tolerated, but functional data from the North Star Ambulatory Assessment (NSAA) criteria showed a nonsignificant trend of improvement in younger patients (<6 years) (NCT03703882) [1]. Edasalonexent may prolong ambulation and delay respiratory and cardiac problems when prescribed for very young patients [24]. Unfortunately, Catabasis Pharmaceuticals stopped development of the drug.

Vamorolone, or VBP15, is a 21-aminosteroid or lazaroid, which is a glucocorticoid steroid that has a delta 9,11 modification of the backbone (Table 1) [25]. Vamorolone stabilizes membranes by reducing lipid peroxidation and inhibiting NF-kB-mediated inflammation. Importantly, vamorolone limits adverse steroid side effects because it binds the glucocorticoid receptor without activating gene expression via the glucocorticoid response element (GRE) [26,27,28,29]. Clinical trials have proven it to be safe, and it has reduced adverse effects (NCT02760264, NCT02760277, NCT03439670) [30]. A non-randomized trial demonstrated maintenance of muscle strength and function for 30 months (NCT03038399). These patients also had normal growth or acceleration of growth for those that previously had been treated with glucocorticoids [22,23,31]. The successes of vamorolone in DMD patients have prompted an ongoing phase 2 pilot trial of the drug in BMD patients as well (NCT05166109). ReveraGen Biopharma is currently preparing to get approval for this drug in the U.S., EU, and U.K.

**Table 1 biomolecules-13-01536-t001:** Treatment strategies for Duchenne muscular dystrophy.

Target	Treatment	Strategies	Outcomes	^1^C/PC	Refs
Inflammation	Edasalonexent	Salicylic acid and DHA, NF-κB inhibitors	Phase 3—no significant improvement in motor function	C	[24]
	Vamorolone	Corticosteroid analog, NF-κB inhibitor	Phase 2—similar to standard of care with fewer adverse events	C	[22,23,31]
Inflammation and fibrosis	Pamrevlumab	CTGF antibody	Phase 3—did not meet primary endpoint	C	
	PEGSerp-1	uPA/uPAR inhibitor	*mdx/Utrn^−/−^* mice, ↑ fiber diameter ↓ fibrosis	PC	[32]
Calcium homeostasis	BI 749327	TRPC6 antagonist	*mdx/Utrn^−/−^* DKO mice, ↑ lifespan ↓ fibrosis	C/PC	[33]
	Pyr10	TRPC3 antagonist	*mdx* mice, ↓ cardiomyocyte fibrosis	PC	[34]
	S48168 (ARM210)	Stabilize RyR	*mdx* mice, ↑ strength ↓ inflammation	C/PC	[35,36]
	CDN1163	SERCA activator	*mdx* mouse, ↓ muscle loss ↑ strength ↓ fibrosis	PC	[37]
	Alisporivir	Inhibits cyclophilin D	*mdx* mouse, ↓ macrophage ↓ fibrosis ↑ muscle regeneration	PC	[38,39,40]
Oxidative stress	Idebenone	CoQ10 analog, antioxidant	Phase 3—did not meet primary endpoint	C	[41,42]
	Flavocoxid	Plant flavonoids, inhibit COX enzymes	Phase 1/2—antioxidant properties alone, temporary anti-inflammatory effect with steroids	C	[43]
	Urolithin A	Metabolite that activates mitophagy	Double-blind trial, ↑ mitochondrial efficiency ↓ inflammation	C/PC	[44,45,46]

^1^C—clinical; PC—preclinical.

Pamrevlumab is a monoclonal antibody specific for CTGF, which regulates pro-fibrotic pathways and is upregulated in DMD patients along with TGF-β [20]. FibroGen assessed pamrevlumab (FG-3019) in two clinical trials in DMD patients, examining its effect on limb muscles and respiratory function. Their phase 3 trial on upper limb performance in non-ambulatory DMD patients found that while pamrevlumab was safe and tolerated, it did not meet its primary endpoint (NCT04371666), whereas the phase 2 trial on respiratory function is ongoing (NCT02606136).

In response to muscle injury, the urokinase-type plasminogen activator and its receptor (uPA/uPAR) form a complex that leads to ECM breakdown, the release of TGF-β, and subsequent macrophage invasion, all necessary processes in the course of normal muscle repair. Normally, active TGF-β is limited by the feedback of plasminogen activator inhibitor 1 (PAI-1) on uPA [47,48,49]. In the context of dystrophic muscle, this system is dysregulated, resulting in highly elevated levels of TGF-β that contribute to fibrosis and macrophage infiltration [50,51]. It was proposed that the chronic inflammation of DMD is linked to persistent uPA/uPAR activity and disruption of the negative feedback loop between TGF-β and PAI-1 [32]. Serp-1 is a protein derived from myxoma virus that mimics the activity of PAI-1, inhibiting uPA and reducing TGF-β. In preclinical studies, a pegylated Serp-1, PEGSerp-1, was administered to *Mdx/Utrn*^−/−^ (DKO) mice, whose phenotype recapitulates the time course and pathology of human DMD [32]. In DKO diaphragms, PEGSerp-1 treatment resulted in increased fiber diameters, decreased fibrotic area, and decreased proinflammatory macrophage invasion [32]. This demonstrated that inhibiting the uPA/uPAR/TGF-β pathway will potentially reduce fibrosis and inflammation. Serp-1 represents a new biological approach for advances in therapeutics to address inflammation and fibrosis.

Since TGF-β has a multifactorial role in signaling during regeneration, therapeutic strategies focused on limiting fibrosis often target this protein. One such strategy is a monoclonal antibody that targets latent transforming growth factor beta binding protein 4 (LTBP4), a matrix-embedded protein that sequesters TGF-β in the ECM and modulates its availability. LTBP4 also binds other members of the TGF-β superfamily, including myostatin, which may aid its efficacy [52]. The antibody is directed against the hinge region of LTBP4; when bound, it maintains the protein in a closed conformation, securing TGF-β and reducing the amount that is freely acting in muscle. Treatment of *mdx* mice for 6 months resulted in increased fiber size and less fibrosis, and there was a synergistic effect when combined with corticosteroids [52]. This preclinical study is encouraging for DMD patients, most of whom are treated with corticosteroids.

### 2.2. Calcium Homeostasis

Disruption of Ca^2+^ homeostasis in muscular dystrophy has been identified as a key contributor to skeletal myofiber and myocardial cell death [53,54,55]. Elevated cytosolic Ca^2+^ activates calcium-sensitive proteases, calpain and phospholipase A2, in myofibers, resulting in proteolysis and membrane damage. Though initially attributed to Ca^2+^ influx through microtears of the sarcolemma, it is clear that increased Ca^2+^ in the cytosol is caused by activation of channels in the membrane of the sarcoplasmic reticulum (SR), such as the ryanodine receptors (RyR1 and RyR2) and the sarcolemma, such as the transient receptor potential canonical channels (TRPC), voltage-gated Na^+^ channels (Na_v_1.4 and Na_v_1.5), and ionotropic purinoreceptors (P2 × 7). Further, in DMD, the reduced activity of the SERCA1 and SERCA2 channels that mediate Ca^2+^ uptake in the SR and the mitochondrial Ca^2+^ uniporter (MCU) also contributes to the loss of calcium homeostasis [56,57]. When cytosolic Ca^2+^ levels were normalized in DMD mouse models, it resulted in reduced fibrosis and muscle pathology while increasing muscle strength [55,58,59]. This predicts that Ca^2+^ channels represent viable therapeutic targets for DMD to prevent triggering chronic inflammation and fibrosis. 

It has been hypothesized that the DGC serves as a scaffold for the assembly of the channels and facilitates the activation of mechanosensitive channels. Ion channels that promote Ca^2+^ influx through the sarcolemma cluster with the DGC by direct interactions with α-syntrophins [60]. In the absence of dystrophin, the integrity of many channels is disrupted, including TRPCs, store-operated Ca^2+^ entry (SOCE) channels, and voltage-gated Na^+^ channels, leading to an excessive influx of Ca^2+^ from the environment. Dystrophic muscle in mice, rats, and DMD patients overexpress TRPC1, TRPC3, and TRPC6, further increasing the Ca^2+^ influx into the muscle [34,61]. The activity of TRPC3 and TRPC6 is increased by phosphorylation by Ca^2+^ calmodulin-dependent kinase (CaMKII) and ERK1/2 and reduced N-linked glycosylation [33,62,63]. The importance of TRPC6 in DMD fibrosis and muscle weakening was demonstrated by treating *Mdx/Utrn*^−/−^ mice, a severe model of DMD, orally with the TRPC6 antagonist BI 749327 [64]. Reduced TRPC6 activity in the double knock-out mice resulted in an extended lifespan, increased muscle and heart function, and reduced fibrosis in the gastrocnemius, diaphragm, and ventricle, as compared to untreated controls. Similar anti-fibrotic outcomes have been observed in mice for cardiac and renal disease [35]. Clinical trials to evaluate the efficacy of BI 749327 in treating kidney disease are ongoing (NCT04176536), raising the possibility that trials with DMD patients could be on the horizon. Similarly, the reduction in TRPC3 activity by treatment with the TRPC3-specific inhibitor Pyr10 reduces fibrosis in cardiac muscle [61]. These data demonstrate the therapeutic potential of TRPC antagonists for reducing inflammation and fibrosis in DMD patients. 

Ryanodine receptors (RyR1-3) are a family of homotetrameric proteins located in the SR that release intracellular Ca^2+^ stores into the cytoplasm in response to membrane depolarization. It has been proposed that Ca^2+^ leakage into the cytosol through RyRs is due to the depletion of the co-factor calstabin-1 [36]. An investigation of potential small molecules able to stabilize calstabin-1 identified S48168(ARM210), which is able to improve muscle strength and reduce pathology in *mdx* mice [36,65]. A phase I trial (NCT04141670) examining the therapeutic potential of S48168(ARM210) to modify RyR1 leakiness in RyR1-related myopathy is currently underway. Based on the *mdx* study, it may act similarly in DMD patients.

The level of cytosolic Ca^2+^ is affected by channels that pump Ca^2+^ back into organelles, including the SR and mitochondria. Perhaps the most important among these is the SR Ca^2+^ ATPase family (SERCA1 and 2), which can remove >70% of the Ca^2+^ from the muscle cytosol during muscle fiber relaxation [37]. In mouse MD models, SERCA activity is decreased due to the upregulation of an inhibitor, sarcolipin. Consistent with this, overexpression of a *SERCA1* transgene or the knockdown of sarcolipin rescued the dystrophic tissue pathology in skeletal muscle and reduced myocardial fibrosis in *mdx* and *Dmd/Utrn^−/−^* mice [57,58,59]. Treatment of *mdx* mice with the SERCA activator CDN1163 for 7 weeks reduced muscular degeneration as measured by treadmill and grip strength and fibrosis in the diaphragm [66]. Though CDN1163 remains in the preclinical stage, it represents a viable therapeutic option for future clinical trials.

The abnormally high intracellular Ca^2+^ concentrations and oxidative stress in the muscle of DMD patients lead to calcium overload of the mitochondria [67] and the opening of the mitochondrial permeability transition pore (MPTP). The open MPTP releases Ca^2+^ and ROS into the cytosol and increases oxidative stress and inflammation [68]. MPTP activation permeabilizes the inner mitochondrial membrane and induces mitochondrial swelling and membrane rupture, causing a loss of the proton gradient and necrotic cell death [68]. MPTP-dependent cell death contributes to the inflammation and pathophysiology of DMD and other MDs. Excess Ca^2+^ in the mitochondria is released when the MPTP opens, further disrupting calcium homeostasis.

Alisporivir, also known as Debio-025, is a non-immunosuppressive analog of cyclosporin A (CsA) that inhibits the opening of the MPTP. This occurs by targeting cyclophilin D, one of the two proteins that make up MPTP [68,69]. Alisporivir was found to be effective in muscular dystrophy in *mdx* and δ-sarcoglycan^−/−^ mice [38,70]. This drug is more effective than prednisone in decreasing muscle fibrosis and infiltration of macrophages and improving muscle regeneration in *mdx* mice [39]. However, this drug suppressed mitochondrial biogenesis and altered the dynamics of other organelles in *mdx* and wildtype mice [40,71]. Alisporivir is currently being investigated as a therapeutic for the treatment of hepatitis C [72], but the mouse data indicate the potential for future therapeutic use in multiple types of MD.

### 2.3. Inhibition of Oxidative Stress

Oxidative stress causes dystrophic pathology by damaging tissue and interfering with repair mechanisms. High levels of reactive oxygen species (ROS) in tissues and cells damage proteins and DNA, induce lipid peroxidation reactions, and activate NF-κB, perpetuating the inflammatory state [41]. ROS levels are elevated in dystrophic muscle due to constant neutrophil invasion, increased activity of NADPH oxidase, and dissociation of nitric oxide synthase from the broken dystroglycan complex. Mitochondria also serve as a tightly regulated source of ROS but are dysfunctional in DMD due to the elevated Ca^2+^ levels due to the damaged sarcolemma. There is a strong interest in compounds with antioxidant properties as potential therapeutics for DMD.

Idebenone is a benzoquinone analog to coenzyme Q10 (CoQ10), which has a similar redox potential as CoQ10, meaning it can scavenge ROS in cells and tissues (Table 1). It also increases ATP production, which is necessary for mitochondrial function, reduces free radicals, and inhibits lipid peroxidation, thus protecting membranes and mitochondria from oxidative damage. In a randomized controlled trial in DMD boys, idebenone demonstrated a trend of improved cardiovascular and respiratory function, but it was not significant [42]. A follow-up study found that this drug significantly improved respiratory function in those DMD patients who were not also taking glucocorticoid steroids [43]. Santhera Pharmaceuticals conducted another phase 3 trial on DMD patients concurrently taking glucocorticoids (NCT02814019) that was terminated due to lack of efficacy.

Another potential antioxidant/anti-inflammatory therapeutic is flavocoxid, a blend of two plant flavonoids, baicalin and catechin (Table 1). It has antioxidant and anti-inflammatory functions. It inhibits cyclooxygenase (COX) and lipoxygenase (LOX) enzymes that convert arachidonic acid to prostaglandins. These, in turn, promote inflammation by activating NF-κB and TNF-α. Prostaglandin levels are elevated in the muscle of DMD patients. Flavonoids are known to inhibit the proinflammatory activity of the NF-κB, TNF-α, and MAPK pathways as well [44]. A phase 1/2 proof-of-concept study in boys with DMD (aged 4–12 years) found it provided a temporary anti-inflammatory effect when given with steroids. Inflammation markers, IL-1β and TNF-α, were reduced at 6 months but returned to baseline at 12 months. Reduced markers of oxidation, specifically peroxide and glutathione peroxidase, were found in the flavocoxid-only group, corroborating the antioxidant effect. Unfortunately, while flavocoxid improved mobility in both younger and older boys, it was recalled by the FDA due to reversible, non-lethal liver issues [44].

Normally, damaged mitochondria are targeted for mitophagy; however, in DMD cardiac and skeletal muscle, this process is impaired, resulting in the accumulation of damaged mitochondria that release high amounts of ROS, mtDNA, and cardiolipin, increasing oxidative stress and inflammation. These molecules activate several inflammatory pathways through IL-1β, the NF-kB signaling pathway, and the inflammasome [67]. Urolithin A, a metabolite that activates mitophagy, has been shown to increase muscle respiration and performance and decrease plasma markers of inflammation in two clinical trials in middle-aged adults (NCT03283462; NCT03464500) [45,46]. Similarly, *mdx* mice treated with urolithin A had fewer damaged mitochondria, reduced fibrosis in the heart and diaphragm, and enhanced regeneration of *mdx* MuSCs. The urolithin A-treated mice also showed functional improvements in grip strength and running endurance [73]. These data indicate that urolithin A could improve the function of and reduce inflammation in DMD muscle.

MSS51 is a mitochondria-associated protein specific to skeletal muscle whose expression is decreased when myostatin is inhibited, suggesting a role in myostatin signaling. *Mss51^−/−^* mice were resistant to diet-induced weight gain and showed increased glycolysis, β-oxidation, and oxidative phosphorylation, indicating MSS51 has a role in the regulation of skeletal muscle mitochondrial respiration and glucose and fatty acid metabolism [74]. When *Mss51* was knocked out in *mdx* mice, they demonstrated larger myofibers, less fibrosis, and greater mitochondrial respiration. These mice showed improvement in running endurance but not in grip strength [74]. These data open the possibility of an investigation into the axis between myostatin, oxidation, and metabolism as an avenue for treatment.

## 3. Emery–Dreifuss Muscular Dystrophy (EDMD)

Emery–Dreifuss muscular dystrophy (EDMD) is a neuromuscular disorder characterized by a combination of skeletal muscle weakness, joint contractures, and cardiac abnormalities [75]. EDMD is rare; it has a prevalence estimated at 3 to 4 cases per million. Falling under the spectrum of laminopathies, EDMD is caused by mutations in at least 10 genes encoding nuclear envelope proteins. We will focus on the most common types caused by mutations in emerin (*EMD*) and lamin A (*LMNA*) [75,76,77]. *EMD* has an X-linked recessive inheritance [77], it encodes emerin, a nuclear membrane protein found at the interface of the nucleus with the cytoplasm. *LMNA* is inherited in an autosomal dominant fashion, and it codes for lamin A and C, which are type V intermediate filaments. The lamins polymerize under the inner nuclear membrane to form the nuclear lamina [78,79].

EDMD patients have diverse clinical presentations, making it challenging to diagnose and manage [75,77,80]. Patients with EDMD have three main sets of symptoms, which consist of early contractures, progressive muscle weakness and atrophy, and cardiac abnormalities. Joint contractures are a hallmark symptom of EDMD, causing multi-joint stiffness, reduced mobility, and functional impairment, most often involving the neck, elbows, and heel. Spinal rigidity may also affect posture, flexibility, and swallowing [75,77]. Progressive muscle weakness and wasting primarily affect the scapulo-humero-peroneal muscles in children, which impairs mobility and strength [75]. In patients with *EMD* mutations, muscle weakness will appear in childhood and progress during adolescence. Cardiac complications will appear after muscle weakness and can include sinus bradycardia, atrioventricular blocks, and paroxysmal atrial fibrillation or flutter, potentially leading to syncope or sudden death [75]. Patients with *LMNA* mutations have a more severe disease with an unpredictable course. These patients have a high risk of ventricular arrhythmias and thromboembolic complications, particularly strokes [75,80].

In the nucleus, emerin binds to lamin A, actin, and nuclear myosin, and as a complex, it regulates chromatin dynamics, acts as a mechanosensor for the nucleus, and participates in the regulation of gene expression [81,82,83,84]. EDMD pathophysiology involves disruptions in the nuclear envelope, resulting in loss of nuclear membrane integrity, heterochromatin decondensation, nucleoplasm leakage, chromatin detachment from the nuclear lamina, pseudoinclusions, and nuclear fragmentation (Figure 1) [75]. These defects contribute to cellular dysfunction, especially in tissues like cardiac and skeletal muscles that undergo mechanical stress [77]. Histologically, the heart demonstrates myocardial fibrosis and fibrolipomatosis, and skeletal muscle has dystrophic changes, including variation in fiber size, increased number of myofibers with centrally localized nuclei, increased endomysial connective tissue, and the presence of necrotic fibers [77]. Similar to DMD, fibrosis in EDMD results from chronic inflammation and mechanical stress on the heart, which promotes fibrotic tissue remodeling [85]. In EDMD-related cardiomyopathy, fibrotic remodeling and adipose deposition in the normal myocardium contribute to the heart defects noted above [85]. Fibrosis significantly impacts EDMD-related cardiomyopathy, so treatment strategies focused on its reduction may reduce cardiac complications.

### EDMD Treatment

EDMD is a challenging and genetically complex disorder, necessitating ongoing research to advance our understanding of the disease and develop effective therapeutic strategies [77,85]. Timely cardiac interventions, regular monitoring, and supportive care are also essential in improving the prognosis and quality of life for individuals with EDMD [75]. Currently, the therapeutic approaches for EDMD patients are limited to the alleviation of symptoms, slowing disease progression, and improving the overall quality of life (Table 2) [75]. Much like DMD and BMD patients, glucocorticoids are prescribed to limit inflammation [76]. Due to side effects associated with long-term use of glucocorticoids, anti-inflammatory and anti-fibrotic agents are being examined in mouse models for EDMD.

Mice deficient in emerin *Emd^−/−^,* display impaired muscle regeneration, inflammation, and fibrotic deposition. Satellite cells (MuSCs) isolated from these mice have defects that cause the failure to regenerate. Emerin binds to and activates histone deacetylase 3 (HDAC3) [86]. MuSCs from emerin-deficient mice treated with theophylline, an HDAC3-specific activator, rescued myotube formation [86]. Similarly, the nuclear membrane defects alter gene expression, and the ERK/MAPK pathway, known to participate in activating inflammation, is upregulated in LMNA cardiomyopathies. PD098059 is a small molecular inhibitor of MEK1 activation of MAPK. When given to *Lmna^H222P/H222P^* mice that lack lamin A/C due to a missense mutation [87], it delayed left ventricular dilatation and improved cardiac function. Similar results were found for the MEK inhibitor selumetinib, which is approved for the treatment of children with neurofibromatosis (NCT01362803) [88]. p38MAPK activation is elevated in LMNA cardiomyopathy, and a phase 2 trial of ARRY-371797, a small molecule p38MAPK inhibitor, as a therapeutic for LMNA cardiomyopathy has been completed (NCT02057341) [88], but no data are available.

A conditional knockout of a muscle-specific nuclear envelope protein, Net39 (Net39 cKO), resulted in a phenotype that recapitulated EDMD; importantly, myoblasts from these mice were hypersensitive to mechanical stretch and had stretch-induced DNA damage [93]. The *Lmna* DK32 mouse model has a very severe form of EDMD, but the symptoms are highly similar to those of the *Net39* cKO mice. When these mice were treated with AAV-Net39, it ameliorated the EDMD symptoms but was unable to completely rescue the phenotype. These data indicate that stabilization of the nuclear envelope may be a viable approach to therapeutics. 

mTOR is a serine/threonine kinase, and it has a role in regulating the production of proteins, lipids, and nucleotides, as well as downregulating autophagy and increasing cellular proliferation and survival [94]. Several rapamycin derivatives, such as temsirolimus, are inhibitors of mTORC1 that are FDA approved for cancer treatment. *Lmna^H222P/H222P^* mice were treated for 14 weeks with temsirolimus, and they demonstrated reactivated autophagy and reduced left ventricular dilatation. Unfortunately, this drug had no effect on cardiac fibrosis [88]. These data indicate that rapamycin and its derivatives may be effective treatments for LMNA cardiomyopathy. 

Gap junctions are important for electrical conduction in the heart and are found on the intercalated disks. Connexin43 (Cx43), the gap junction protein, has altered expression in *Lmna^H222P/H222P^* mice due to an altered microtubule cytoskeleton [89]. Paclitaxel is a taxane chemotherapeutic that binds to tubulin and inhibits microtubule disassembly, thereby inhibiting cell division. When *Lmna^H222P/H222P^* mice were treated with paclitaxel, Cx43 was once again found in the intercalated discs [89]. Paclitaxel potentially can stabilize conduction, reducing the severity of LMNA cardiac defects. 

Continued research efforts and collaborative approaches are crucial to advancing our understanding of EDMD and developing effective, innovative therapies that target fibrosis and improve EDMD prognosis [85,93]. Recent advancements in omics technologies have improved prognostic predictions regarding potential therapeutic targets [95]. 

## 4. Facioscapulohumeral Muscular Dystrophy (FSHD)

Facioscapulohumeral muscular dystrophy (FSHD) is the third most prevalent type of MD, following myotonic dystrophy type 1 and DMD, estimated to affect 1 in 15,000 to 1 in 20,000 people [96]. FSHD is inherited in an autosomal dominant manner; however, approximately 30% are de novo cases. Additionally, there is a high frequency of somatic mosaicism [97]. FSHD typically presents with weakness in one or more of these facial muscles, the stabilizers of the scapula, or the dorsiflexors of the foot. Muscle weakness is slowly progressive and, with time, can involve the axial and respiratory muscles and those of the lower limbs [98]. The severity of the disease course is variable; as many as 20% of affected individuals will require a wheelchair. Symptoms typically appear in the second or third decade of life but can appear in children [99]. Life expectancy is not shortened [98,100].

Histopathological studies of human FSHD biopsies have shown that there are regenerating myofibers of varying sizes; inflammation and fibrosis that is both endomysial and perivascular were prominent, and there was significant fat replacement of the muscle fibers [101]. A two-year prospective study of FSHD patients found that inflammation was mild in the early stages, but its presence was related to an increased rate of muscle degradation and fat infiltration [102]. When examining RNAseq and microarray data from FSHD biopsies, it was noted that inflammatory genes have increased levels of expression [103]. FSHD myoblasts have increased sensitivity to oxidative stress and are prone to apoptosis because double homeobox protein 4 (DUX4) affects mitochondrial function. As the disease progresses, regeneration becomes increasingly impaired, further expanding the level of adipose tissue in the muscle [103].

*DUX4* is the gene affected in FSHD patients [104,105]. *DUX4* is expressed in pre-implantation embryos, and Dux binding sites are found in the control regions of genes involved in early genome activation [103,104]. DUX4 is important in early embryos for embryonic genome activation (EGA). DUX4 triggers this process and subsequently becomes inactive [106,107,108,109,110]. Postnatally, DUX4 is expressed at low levels in the testis and the thymus. Ectopic expression in skeletal muscle postnatally results in FSHD [111,112,113]. The human *DUX4* gene is located on chromosome 4 in a region known as D4Z4, which consists of 11 to more than 100 repeated 3300 DNA base pair segments. Each of the repeated segments in the D4Z4 region contains a copy of the *DUX4* gene; the copy closest to the end of chromosome 4 is called *DUX4*, while the other copies are referred to as “DUX4-like” or *DUX4L* [114,115]. Typically, people have 11–100 D4Z4 repeats present in heterochromatin, resulting in a lack of transcription [101]. There are two forms of FSHD; the most common form, FSHD1, in approximately 95% of patients, occurs in people that have less than 10 D4Z4 repeats. This repeat contracture results in hypomethylation, causing *DUX4* to be aberrantly expressed in skeletal muscle (Figure 1) [116,117,118]. FSHD2 is rare, and D4Z4 derepression is caused by defects in D4Z4 chromatin repressors, most often SMCHD1 [117,119,120]. The focus of research into treatment for FSHD currently is focused on improving disease outcomes. 

### Small Molecule Therapies

Studies in both animals and humans suggest that β2-adrenergic agonists can stimulate muscle growth and prevent atrophy following various types of injuries (Table 2) [121]. Two non-targeted trials of β2-adrenergic agonists in FSHD patients demonstrated limited beneficial effects on muscle mass and strength in the short term [90,91]. More recently, a targeted screen of a chemical library enriched for compounds with epigenetic activity implicated β2 adrenergic agonists, such as albuterol, and the bromodomain and extra-terminal domain (BET) protein family member BRD4, as inhibitors of *DUX4* expression [122]. The BRD4 inhibitor blocks the binding of acetylated histones and transcription factors and activation of RNA polymerase II. Conversely, β2 adrenergic receptor agonists suppress *DUX4* expression by activating adenylate cyclase to increase cAMP levels [122]. MAPK is activated by β2 adrenergic signaling, and this screen identified p38α and p38β MAPK isoforms as potent inhibitors of DUX4 [113]. Consistently, the MAPK inhibitor losmapimod, when used in a mouse xenograft model, reduced DUX4-driven gene expression [123]. Losmapimod was tested in a randomized, double-blind phase 2b clinical trial (NCT04003974) with FSHD1 patients [92]. While there was no decrease in the expression of DUX4-driven genes, treatment did reduce the fatty replacement in muscle [92]. The positive effects of losmapimod may be attributed to its anti-inflammatory properties rather than its ability to reduce DUX4 levels [113]. 

To induce transcription of target genes, DUX4 recruits the histone acetyltransferases p300 and CREB to target loci, leading to H3K27 acetylation [124]. Therefore, blocking the transcription of certain genes may be a possible avenue for new FSHD therapeutics, and some data indicate that small molecule inhibitors are promising options. Treatment with a novel p300-specific spirocyclic HAT inhibitor, iP300w, effectively prevented the induction of DUX4 target genes, inhibited the harmful effects of DUX4, and reversed the accumulation of acetylated histone H3 in C2C12 cells and iDUX4pA mouse model of FSHD [125]. 

## 5. Limb-Girdle Muscular Dystrophy

Limb-girdle muscular dystrophy (LGMD) is the largest group of muscular dystrophies. They were originally defined by the progressive wasting of skeletal muscles of the pelvic and pectoral girdles. These dystrophies show significant variation in the onset of the disease, degree of wasting and inclusion of cardiomyopathy, cardiac arrhythmias, and respiratory failure [126]. There are 24 genetic subtypes based on disease phenotype and mutations: (i) sarcoglycan complex (LGMD2C-F); (ii) glycosylation/α-dystroglycan complex (LGMD2I, LGMD2K, LGMD2M, LGMD2N, LGMD2O, LGMD2P, LGMD2S, LGMD2T, LGMD2U, LGMD2Z); (iii) sarcomeric proteins (LGMD1A, LGMD1D, LGMD1E, LGMD2A, LGMD2G, LGMD2J, LGMD2Q, and LGMD2R); (iv) signal transduction (LGMD1C, LGMD2P, LGMD2W); and (v) membrane trafficking and repair (LGMD1C, LGMD1F, LGMD2B, LGMD2L). These disorders are relatively rare compared to DMD, with individual estimated prevalences of 0.01–0.60 cases per 100,000 persons [127]. The rarity of these disorders limits the ability to carry out clinical trials to examine the effectiveness of therapeutic strategies discovered in mice [128]. Below, we describe LGMD subtypes due to mutations in sarcoglycans, dystroglycan, and dysferlin and the current status of anti-inflammatory/anti-fibrotic strategies for these LGMD subtypes (summarized in Table 3). 

### 5.1. Sarcoglycanopathy

Sarcoglycan consists of four subunits, α, β, γ, and δ, that form an integral membrane protein complex that is a subunit of the DGC in skeletal muscle. This protein complex is essential for the structural stability of the sarcolemma in both skeletal and cardiac muscle (Figure 2). Genetic mutations of any sarcoglycan subunit are inherited in an autosomal dominant fashion and result in sarcoglycanopathy, which has a similar phenotype to DMD. In sarcoglycanopathy, or LGMD2, the average age at onset of muscle weakness is 6–8 years, with loss of ambulation at 12–16 years old. Similar to DMD, cardiomyopathy and respiratory complications following the loss of ambulation are commonly observed [126,129,130]. At the tissue level, muscle weakness is associated with chronic inflammation and the replacement of muscle fibers with fat depositions and fibrosis [131].

Because of the phenotypic similarity to DMD, anti-inflammatory corticosteroids are prescribed, but their therapeutic benefits in the management of sarcoglycanopathies have been difficult to evaluate due to the rarity of these patients. However, there are case reports of LGMD2D (α-sarcoglycan) and LGMD2E (β-sarcoglycan) patients that describe the benefits of deflazacort and prednisone treatment in these patients [132,133]. 

A more direct approach to regulating fibrosis is to target the fibroblasts and fibroadipogenic progenitor cells (FAPs) at the site of injury. Since these cells are the primary source of fibrotic scarring associated with muscular dystrophy, disrupting the FAPs represents a cogent approach to reducing fibrosis. Nintedanib, a tyrosine kinase inhibitor, disrupts signaling through the receptors for platelet-derived growth factors (PDGF), fibroblast growth factor (FGF), and vascular endothelial growth factor (VEGF) and disrupts the fibrotic activity of muscle fibroblasts isolated from mdx mice [134,135]. Treatment of a murine model of α-sarcoglycan-deficient mouse (*Sgca^−/−^*) with nintedanib resulted in increased muscle function based on distance on a treadmill and grip strength as compared to untreated controls. An examination of the quadriceps, gastrocnemius, and triceps revealed reduced inflammation and fibrosis and expression of cytokines associated with these processes (e.g., *Tgfb1, Tnf*, and *Nfkb1*) [136].

**Table 3 biomolecules-13-01536-t003:** Treatment strategies for limb-girdle muscular dystrophies.

	Treatment	Strategies	Outcomes	^1^C/PC	Refs
Sarcoglycanopathies					
	Nintedanib	Tyrosine kinase inhibitor	↑ muscle function in *Sgca^−/−^* mice	PC	[136,137]
	Metformin	Metabolic regulator acting through AMPK	↓ fibrosis in diaphragm and heart of *Sgcd^−/−^* mice	PC	[138]
	A438079	P2X7 antagonist	* Sgca^−/−^ * mice ↓ fibrosis, ↓ inflammatory cell infiltration	PC	[139]
Dystroglycanopathies					
	Tamoxifen and raloxifene	Inhibits the Akt/mTor pathway	↓ fibrosis, ↓ inflammation, ↓ muscle damage, ↑ muscle fiber size	PC	[140]
	BBP-418 (Ribitol)	Sugar substrate glycotransferase	↓ muscle breakdown and ↑ functional measures in LGMD2I patients	PC	[141]
Dysferlinopathies					
	Vamorolone	Corticosteroid analog, NF-κB inhibitor	↓ lipid mobility, ↑ muscle repair, ↓ fatty deposits in Dysf^−/−^ mice	PC	[30,142]
	Galectin-1 (Gal-1)	↑ laminin and α7β1 integrin binding	↓ inflammation and fat deposition, ↑ membrane repair in Dysf^−/−^ mice	PC	[143]
	Acid sphingomyelinase	Ceramide production to promote sarcolemma repair	AAV vector delivery in Dysf^−/−^ mice ↑ membrane repair, ↓ fatty deposition, ↑ myofiber size and muscle strength	PC	[144]

^1^C—clinical; PC—preclinical.

An alternative therapeutic approach for sarcoglycanopathy patients is treatment with metformin. This antidiabetic drug acts as a metabolic regulator and promotes autophagy through the AMP-activated protein kinase (AMPK) [137]. In mice deficient in δ-sarcoglycan (*Scgd^−/−^)* that recapitulate the aggressive pathology of LGMD2F, 4 weeks of treatment was sufficient to reduce fibrosis in the diaphragm as compared to untreated *Scgd^−/−^* mice. Further, there was reduced myocardial fibrosis and cardiomyocyte hypertrophy, indicating it has cardioprotective properties [145]. Metformin has also been shown to improve sarcolemma integrity in *mdx* mice as measured by the amplitude of miniature endplate potentials [146]. Phase III clinical trials with metformin and the nitric oxide precursor 1-citrulline in DMD patients resulted in positive but not statistically significant changes in motor function in the treated cohort [138].

A third approach to reducing fibrosis and inflammation in sarcoglycanopathies focuses on the P2X7 ionotropic receptor that triggers the inflammasome in response to binding ATP in the extracellular space [147]. The extracellular surface of the sarcoglycan complex possesses ATPase activity; thus, when sarcoglycan-deficient muscle fibers lose their sarcolemmal integrity, they will have higher levels of extracellular ATP to trigger P2 × 7 [148]. Recent studies in *Sgca^−/−^* mice have shown that the P2X7 antagonist, A438079, reduces fibrosis and the infiltration of innate inflammatory cells into the gastrocnemius, quadriceps, and anterior tibialis muscles [149]. The immunomodulatory effect of the P2X7 antagonist makes it ideal for limiting muscle fiber damage from proinflammatory macrophages and neutrophils.

### 5.2. Dystroglycanopathies

Dystroglycanopathies are autosomal recessive dystrophic disorders with variability in the age of onset and degree of muscle wasting. α- and β-dystroglycan form an integral membrane protein complex that links the basement membrane to the cytoplasmic dystrophin protein. α-dystroglycan binds to ECM and synaptic proteins and is dependent on heavy glycosylation [150] (Figure 2). Dystroglycanopathies are primarily linked mutations in glycosyltransferases that target α-dystroglycan, including fukutin-related protein (*FKRP*/LGMD2I) [139], fukutin (*FKTN*/LGMD2M), -like-acetylglucosaminyl transferase (*LARGE*), protein-O-mannosyltransferase 1 (*POMT1*/LGMD2K), protein-O-mannosyl transferase 2 (*POMT2*/LGMD2N), protein-O-mannose-1,2-N-acetylglucosaminyltransferase 1 (*POMGNT1*/LGMD2O), dolichyl-phosphate mannosyltransferase polypeptide 3 (*DPM3*), and isoprenoid synthase domain containing (*ISPD*/LGMD2U) [151,152,153,154,155,156]. A rare subset of dystroglycanopathies includes mutations in the dystroglycan gene (*DAG1*). These mutations cluster in the N-terminal domain of α-dystroglycan or in the domains that stabilize the α/β-dystroglycan interface [157]. Though these milder forms of muscular dystrophy are classified as LGMD based on their progressive muscle wasting in the extremities, more severe congenital muscular dystrophies are associated with mutations in glycosyltransferases targeting α-dystroglycan that result in a non-progressive muscle weakness that is observed at birth (discussed in Section 6). Most notable among these are Walker–Warburg (WWS) and muscle eye brain syndromes (MEB), which present with an earlier onset of muscle weakness and include complex eye and brain disorders [158,159]. Our focus will be on the milder LGMD forms.

The most common form of dystroglycanopathy, LGMD2I, is caused by mutations in *FKRP*; it occurs at a frequency of 1 in 250,000 in the U.S. The use of corticosteroids to reduce fibrosis and muscle loss has shown promise with LGMD2I. Mice that recapitulate the LGMD2I phenotype through mutation of *FKRP* [156] have reduced pathology and muscle degeneration when treated with prednisolone, though it did not improve muscle strength. However, in combination with the bisphosphonate, alendronate, which is normally administered to limit osteoporosis, the level of muscle damage was lessened, and muscle function was improved [155]. Under both conditions, the level of glycosylated α-dystroglycan increased, suggesting a mechanism other than immunosuppression for injury reduction. 

Other immunosuppressive agents possessing anti-fibrotic activity have been tested for their therapeutic potential in dystroglycanopathies. Among these, the estrogen receptor modulators tamoxifen and raloxifene have been used for long-term (>1 year) treatment of *FKRP*-deficient mice [158]. Marked improvement in muscle strength, as judged by grip power, running distance, and duration, were observed, as well as improved cardiac and respiratory function. The use of rapamycin also has been explored as a treatment for LGMDM2, dystroglycanopathies caused by mutations in *FKTN*. Rapamycin is an inhibitor of the Akt/mTOR signaling pathway that inhibits muscle regeneration and repair and promotes fibrosis [159,160]. In mice with a conditional muscle-specific knockout of the *Fktn* gene, a 4-week course of rapamycin was sufficient to reduce fibrosis, inflammation, and muscle damage while increasing muscle fiber size [161]. The collagen matrix associated with fibrosis is derived from fibroblasts, suggesting that muscle is not the sole target for rapamycin. 

An alternate pharmacotherapeutic approach that has seen considerable success in improving dystroglycanopathy is treatment with BBP-418 (Ribitol). This sugar is a substrate in the enzymatic cascade, leading to α-dystroglycan glycosylation. By increasing the cellular substrate concentration, it is proposed that less efficient mutated glycotransferases will be able to modify α-dystroglycan [140]. Ongoing open-label phase 2 trials (NCT05775848, NCT04800874) in ambulatory and non-ambulatory LGMD2I patients have shown signs of effectiveness. After 12 months, the trial revealed a sustained reduction in creatine kinase in these patients, suggesting a reduction in muscle breakdown. They also demonstrated an improvement in functional measures, including a 10 m walk test [162].

### 5.3. Dysferlinopathy

Dysferlin belongs to the mammalian family of Fer-1-like proteins [141,163]. The dystrophy-related Fer-1-like (*DYSF*) gene has 56 exons, and it has 2 main and 15 known alternative transcripts [164]. Dysferlin is a transmembrane protein, 230 kDa in size, with seven conserved CA2 domains. These domains are approximately 130 amino acids in length and bind to lipids and Ca^2+^, regulating Ca^2+^-mediated signaling events. There are two other domains, FerA and DysF, whose function is unclear [141,165,166]. Dysferlin is found in the sarcolemma, the membranes of cytoplasmic vesicles, and T tubules in muscle fibers [164]. In T tubule membranes, it has a role in the biogenesis and maintenance of the T tubule system [167,168]. Dysferlin is also expressed in membranes in the brain, heart, liver, lungs, and, notably, monocytes and macrophages [169,170].

There are more than 260 polymorphic mutations in DYSF reported to result in three types of muscular dystrophies: limb-girdle autosomal recessive muscular dystrophy type 2B (LGMDR or LGMD2B), Miyoshi myopathy (MM), and distal myopathy with anterior tibial onset (DMAT) (Figure 2) [166]. In Miyoshi myopathy, a distal muscular dystrophy, the calf and plantar muscles are the first to be affected. As the disease progresses, it will involve the hamstrings along with the gluteal muscles [164,171]. The average age of onset is 14–40 in both men and women. In contrast, LGMD2B is a proximal muscular dystrophy, which causes atrophy and weakness of the proximal thigh and pelvic girdle muscles. Some patients have simultaneous weakness of proximal and distal leg muscles, and the shoulder girdle is less often affected [164]. The average age of diagnosis is 25 years of age. DMAT is a rapidly progressing form of dysferlinopathy with initial weakness of the anterior tibial muscles rapidly followed by proximal muscle weakness of both limbs and loss of ambulation within 10 years [172]. Some patients demonstrate cardiac and respiratory deficiencies, but this is rare, and the etiology is unclear as patients with significant limb muscle involvement do not have diaphragm weakness [166,172,173].

The main function of dysferlin is the repair of damage to cell membranes. In normal muscle, the initial response to sarcolemmal damage is an influx of Ca^2+^ (Figure 2). At the same time, there are increased levels of Mitsugumin 53 (MG53), a member of the tripartite motif-containing protein family (TRIM72), at the site of damage [174,175]. Since dysferlin is anchored in both the sarcolemma and membranes of cytoplasmic vesicles, it interacts with MG53 and several other proteins, such as Annexin A2 and AHNAK, to co-ordinate the docking and fusion of the vesicles with the sarcolemma (reviewed in [175]). On biopsy, the different dysferlinopathies share a marked inflammatory response, incomplete muscle regeneration, altered Ca^2+^ signaling, and increased adipogenic replacement [167,168,176,177,178].

Deficient myofiber sarcolemma resealing activates a chronic inflammatory response that results in muscle degeneration in dysferlinopathy. Consistently, dysferlin-deficient mice (AJ) can be rescued by overexpression of dysferlin transgene as judged by histopathology, macrophage infiltration, and muscle function [179]. This was recapitulated by overexpression of a natural shortened “minidysferlin” mutation found in LGMD2B patients [180], suggesting a possible approach for gene therapy. However, the minidysferlin was not able to prevent myofiber degeneration after eccentric activity [181]. 

Macrophages isolated from AJ mice have increased phagocytic activity and release excessive amounts of intracellular vesicles that result in elevated secretion of cytokines and other factors that trigger inflammation [182]. Further, dysferlin-deficient muscles in mice and humans demonstrate high levels of C5b-9 complement deposition in their membranes [178]. Proteins of the inflammasome pathway, as well as proinflammatory cytokines and chemokines, are upregulated and activated in the absence of dysferlin in mice and LGMD2B patients [182,183]. Further, two independent dysferlin mutations in mice (A/J and BLAJ) and human patients demonstrated high levels of fatty replacement of myofibers [168]. This has led researchers to posit that the loss of sarcolemma repair leads to a combination of factors that ultimately result in chronic inflammation and incomplete muscle regeneration.

Membrane instability leads to inflammation in dysferlinopathies similar to DMD; thus, it was logical to consider treating LGMD2B and MM patients with glucocorticoids. But, in a randomized controlled trial, treatment of these patients with the guideline anti-inflammatories for DMD patients [19], deflazacort or prednisone, was not effective and worsened muscle loss and weakness [184]. Interestingly, *Dysf^−/−^* mice that also lacked the complement protein C3 demonstrated amelioration of symptoms, whereas *mdx/C3*^−/−^ mice did not [185]. This predicts inflammation is regulated by a distinctly different mechanism in dysferlinopathies as compared to DMD. Currently, there are no FDA-approved drugs to improve myofiber repair or address inflammation and other disease etiologies in LGMD2B and MM patients.

Vamorolone, or VBP15, is a first-in-class steroidal anti-inflammatory. This drug was found to inhibit NF-kB-mediated inflammation while limiting adverse steroid side effects [26,27,28,29]. Vamorolone increased membrane repair in LGMD2B patient-derived myoblasts. Further, when delivered orally to B6AJ mice, it decreased lipid mobility, increased muscle repair, and reduced fatty replacement of myofibers, all of which were made more severe by prednisone treatment [186]. Currently, vamorolone is in clinical trials as an anti-inflammatory for DMD/BMD, and preliminary findings reported fewer adverse events and some improved motor function [30,187]. While the data are intriguing, there are no trials in LGMD2B or MM patients currently. 

Another treatment approach uses synthetic membrane stabilizers to protect the sarcolemma from mechanical stress. One example is poloxamer 188 (P188), which is FDA approved to reduce blood viscosity before transfusions and is a component of toothpaste, cosmetics, and other pharmaceuticals [142]. P188, also referred to as PLURONIC F68, FLOCOR, and RheothRx, is known to repair damaged cell membranes [142,188,189]. In *mdx* mice, P188 delivered subcutaneously improved the myocardium and limb muscles [190]. Similar findings were reported in dystrophic dogs, where it prevented cardiac dilatation [191]. Its proven safety and efficacy in membrane repair would indicate that P188 is an excellent choice as a therapeutic for dysferlinopathies. Unfortunately, the only clinical trial that was designed to examine P188 as an LGMD2B treatment was discontinued due to lack of funding. 

Galectin-1 (Gal-1) is a soluble, 14kDa, non-glycosylated β-galactosidase binding protein that contains a carbohydrate recognition domain (CRD) [192]. During muscle regeneration, Gal-1 interacts with laminin and α7β1 integrin to modulate myoblast fusion. It also plays a role in the transdifferentiation of fibroblasts into muscle [193,194]. Gal-1-deficient mice have reduced myoblast fusion and impaired regeneration [195]. When recombinant murine Gal-1 was administered to *mdx* mice, they demonstrated improved muscle strength and activity. Unfortunately, there was no significant improvement in muscle pathology [196]. More recently, when recombinant human Gal-1 (rHsGal-1) was administered to BlaJ dysferlin-deficient mice, they demonstrated reduced inflammation and fat deposition and increased membrane repair. Similarly, myotubes derived from patient cells demonstrated increased membrane repair when treated in vitro [143].

When cell membranes are damaged, lysosomes are tethered to the membrane by dysferlin as a step in their repair. Acid sphingomyelinase (ASM) is a lysosomal enzyme that cleaves membrane sphingomyelin, generating ceramide that is secreted from these lysosomes. The presence of ceramide in the outer leaflet of the membrane induces the rapid endocytosis of the damaged region [197]. Dysferlinopathic muscle cannot tether lysosomes to sites of damage in the membrane, causing a delay or loss of ASM secretion, which, in turn, activates the inflammatory response. Bittel et al. [198] delivered the human ASM protein in an AAV vector targeted to the liver in the B6A/J mice and found that this improved membrane repair, decreased fatty deposition, and increased myofiber size and muscle strength. The delivery of this protein to the liver rather than expressing dysferlin itself is an interesting approach, as there are significant barriers to delivering full-length dysferlin via AAV to skeletal muscle [144]. While the liver turnover of cells will require multiple AAV injections over time, it does represent a potential treatment for LGMD2B and MM patients.

## 6. Congenital Muscular Dystrophy

Congenital muscular dystrophies (CMDs) are a heterogeneous group of autosomal recessive disorders distinguished by severe hypotonia and extreme muscle wasting at, or soon after, birth [199]. The muscle pathology is non-progressive, and symptoms can include microcephaly, eye disorders, cerebral malformations, learning disorders, and respiratory insufficiencies [199]. The most prevalent CMDs are caused by a disruption of the interaction between the extracellular matrix and the sarcolemma that is essential for cell membrane stability and cell viability. These dystrophies include LAMA2-related muscular dystrophy (MDC1A), caused by mutations in the α chain of laminin-211 (0.81 in 100,000 individuals worldwide), Ullrich congenital muscular dystrophy (UCMD) (0.13 per 100,000 individuals) and Bethlem myopathy (0.77/100,000 individuals), which are associated with mutations in the α chain of collagen VI (ColVI) (Figure 3) [200,201]. Other CMDs, such as WWS, MEB syndrome, and Fukuyama congenital muscular dystrophy, are the result of deficiencies in the glycosyltransferases that target α-dystroglycan and are essential for laminin binding (see Section 5.2).

### 6.1. LAMA2-Related Muscular Dystrophy

MDC1A patients display hypotonia and muscle wasting at birth and with progressive spinal deformities and contractures of the major limb joints in early childhood [202,203]. These patients can suffer from neuropathies under the age of 1 and can experience cerebral atrophy and seizures in older individuals [204,205,206]. In severe cases, the loss of muscle strength complicates swallowing and breathing, and death due to respiratory insufficiency can occur in the first decade, though their usual life expectancy is the thirties [207,208]. 

Inflammation and fibrosis play prominent roles in the early onset of muscle weakness in MDC1A [209]. Individuals experience a burst of inflammation in their skeletal muscle as early as a couple of months after birth. This leads to the expansion of the interstitial ECM between myofibers, resulting in fibrosis. Unlike DMD, where fibrosis builds over time with repeated rounds of muscle breakdown and repair, fibrosis in MDC1A appears early and is maintained [209]. Several mouse models for a *Lama2* deficiency have been described that include phenotypically moderate spontaneous mutations (e.g., *dy/dy* and *dy^2J^/dy^2J^*) and phenotypically severe targeted null mutations (*dy^W^/dy^W^* and *dy^3K^/dy^3K^*) in the *Lama2* loci [210]. The targeted mutations recapitulate the MDC1A disease course, including muscle wasting, reduced muscle regeneration, inflammation, fibrosis, and a shortened life span [211,212,213]. 

Laminin-211 is an ECM protein comprised of one α2 chain, one β1 chain, and one γ1 chain that acts as a ligand for α-dystroglycan, integrin α7β1, agrin, and fibulins [214,215,216]. Deficiencies of the α2 subunit (LAMA2) disrupt interactions with both receptors. Integrin α7β1 binding to laminin-211 activates the PI3K/Akt and Ras/Raf/MEK/ERK cell survival pathways that signal through focal adhesion kinase (FAK). Disruption of this interaction promotes atrophy and apoptosis [217].

Fibrosis in MDC1A muscle has been linked to the TGF-β and renin-angiotensin signaling pathways. As described in Section 2, TGF-β is a primary regulator of fibrotic deposition via its roles in promoting the expression of ECM proteins and the conversion of satellite cells to a fibroblastic lineage [20,218]. Further, increased TGF-β activity was observed during muscle development in Lama2-deficient mice (*dy^W^/dy^W^*) [211]. Angiotensin II (Ang II) has dual roles in muscle. By binding to the angiotensin I receptor (AT_1_R), it promotes muscle wasting through activation of the ubiquitin-proteasome pathway and fibrosis through the upregulation of TGF-β and its targets, Smad2/3 [219,220,221,222]. Both *Ang II* and *AT_1_R* expression are upregulated in the muscle of DMD, BMD, and MDC1A patients [211,223].

Therapeutic strategies designed to reduce fibrosis in MDC1A patients have largely focused on reducing TGF-β levels (Table 4). One approach that has shown promise is the AT_1_R blocker losartan. Though initially approved by the FDA to control hypertension, it effectively reduced fibrosis in the *dy^W^/dy^W^* and *dy^2J^/dy^2J^* by disruption of Ang II signaling through AT_1_R, which inhibits the conversion of TGF-β to its active form [224,225,226,227]. Losartan treatment also decreased ERK phosphorylation and fibrosis in *dy^W^/dy^W^* mice [224]. As the Ras-Raf-MEK-ERK signaling pathway promotes inflammation when overactivated, this suggests an alternative approach for losartan to prevent fibrotic build-up [228]. Though there is considerable preclinical interest in losartan as an anti-fibrotic agent, there are no ongoing trials for its use in MDC1A.

The FDA-approved serine/threonine kinase inhibitor vemurafenib effectively ameliorates fibrosis in *dy^3K^/dy^3K^* mice [229]. Treatment resulted in a reduction in TGF-β and mTORC1/p70S6K signaling. Much like losartan, vemurafenib was able to block the pro-fibrotic pathway but was unable to promote muscle repair. This raises concerns about its use as a solo therapy. A third anti-TGF-β therapeutic, holofuginone, has been shown to reduce fibrosis in *dy^2J^*/*dy^2J^* mice [230]. Treated mice also experience increased muscle fiber diameter and improved motor coordination, predicting a role in muscle repair. 

In addition to blocking the TGF-β signaling pathway that directly induces fibrosis and inflammation, other research groups have explored the therapeutic benefits of blocking secondary events associated with muscle pathology in MDC1A patients and mouse models. Perhaps the most promise has been reported with omigapil, an inhibitor of the pro-apoptotic protein, Bax, and the GAPDH-Siah apoptosis cascade [231,232]. A 17.5-week treatment of *dy^2J^/dy^2J^* mice led to reduced muscle fibrosis, increased grip strength, and improved respiratory rates [242]. Clinical trials with MDC1A and COL6 congenital muscular dystrophy (COL6-RD) patients have been completed by Santhera Therapeutics to establish the pharmacokinetics of omigapilin in children and adolescents (NCT01805024). Further trials have not been announced by the company.

The pathology in *Lama2*-deficient mice appears to be exacerbated by the upregulation of lysosome-associated degradation pathways in the muscle [233,243]. Consistent with this, inhibition of the ubiquitin-proteasome system by treatment of *dy^3K^/dy^3K^* mice with MG-132 or bortezomib increased muscle strength and mobility and reduced muscle fibrosis while extending the mouse’s lifespan [233,234]. In wildtype mice, Lama2 acts as an inhibitor of autophagy through the inhibition of the class III PI3K Vps34. Thus, *Lama2*-deficient mice, *dy^3K^/dy^3K^*, express higher levels of autophagy markers *Foxo3* and Beclin1 (*BECN1*), suggesting a higher rate of autophagy [233]. Systemic injection of *dy^3K^/dy^3K^* mice with 3-methyladenine (3-MA) that inhibits Vps34 [235] ameliorates muscle pathology [243]. Though these approaches identify potential targets for the development of novel therapeutic approaches for MDC1A, the available drugs act systemically and are likely to have negative side effects in patients [225]. 

Oxidative stress represents a third target for the treatment of secondary events in *Lama2*-deficient skeletal muscle. A study of *dy^2J^/dy^2J^* mice and MDC1A patients has observed significantly elevated levels of reactive oxygen species (ROS) that can contribute to muscle damage [244]. Delivery of N-acetylcysteine (NAC), an antioxidant that scavenges ROS in *dy^2J^/dy^2J^* mice, served to protect against muscle injury that leads to improved muscle strength while reducing inflammation and fibrosis [236]. Similar results were observed for a second antioxidant, vitamin E. Unlike NAC, vitamin E appears to work through the repair of the sarcolemma damage [237]. The outcome of reducing muscle damage and the associated inflammation and fibrosis is the same [236]. This represents a fertile avenue for clinical trials with MDC1A. 

It has become apparent that the pathology of skeletal muscle in MDC1A is dependent on multiple regulatory pathways that contribute to myofiber breakdown and repair, as well as inflammation and fibrosis. This makes MDC1A refractory to therapeutic agents that target a single pathway. Greater success has been seen in mouse models with a combinatorial treatment. Perhaps the best example is combining promyogenic growth factors, such as IGF1, with anti-fibrotic losartan [227]. Similarly, promoting cell stability in the muscle of *dy^W^/dy^W^* mice by the introduction of a mini-agrin transgene has an additive effect when combined with the apoptosis inhibitor omigapil [245]. However, the existing MDC1A therapeutic targets do not address the motor nerve pathology associated with the disease. Additional research is required to develop a strategy to address the neuropathies. 

### 6.2. Collagen VI-Related Dystrophies

ColVI is a beaded fibril consisting of three chains encoded by *COL6A1*, *COL6A2*, and *COL6A3*. The ColVI collagen fibers form a mesh-like network associated with the ECM of basement membranes, where it can modify the nature of the structure based on binding to glycoproteins, proteoglycans, and glycosaminoglycans [246]. They serve important roles in protecting cells from compressive forces while facilitating the interaction between cells and ECM proteins. Further, ColVI participates in the regulation of autophagy through direct interaction with integral membrane proteins on adjacent cells (β1 integrin, TEM8, and VEGFR2) [247,248,249].

Mutations in *COL6A1*, *COL6A2*, and *COL6A3* lead to a spectrum of congenital muscular dystrophies that range from UCMD at the severe end of the spectrum to Bethlem myopathy at the mild end [246]. UCMD patients experience congenital hypotonia and contractures in joints of the extremities. Motor milestones are often delayed, and ambulation is limited. Similar to MDC1A, UCMD patients suffer from respiratory insufficiencies that commonly lead to death in the teenage years [246]. 

The histopathology associated with muscle weakness in ColVI-related dystrophies includes muscle fibers of various sizes and a significant increase in interstitial fibrosis [250]. As with MDC1A, fibrosis accumulation is immediate and does not build with time, as seen with DMD. Using *Col61A*-deficient mice as a model, fibrosis was found to be related to overactivation of mesenchymal stem cells that have been converted to the fibroblast lineage [250].

Studies in Col6a1-deficient mice revealed that COL6A1 is required for the progression of autophagy in muscle. Markers of autophagy, including beclin-1, BCL-2/adenovirus E1B-interacting protein-3 (Bnip3), and the microtubule-associated protein 1A/1B-light chain 3 II (LC3-II), are all reduced in these mice [251]. Further, the sarcoplasmic reticulum and mitochondria have alterations in their ultrastructure that are consistent with reduced autophagy [238]. A consequence of this is the opening of MPTP and Ca^2+^ dysregulation (see Section 2.3), leading to increased apoptosis. This mitochondrial phenotype also was observed in myoblasts isolated from UCMD patients [252]. 

Treatment with cyclosporin A (CsA), an inhibitor of MPTP, rescued the ultrastructure defects and reduced apoptosis (Table 4) [238]. Small open trials treating UCMD patients with CsA for short-term (1 month) or long-term (>1 year) periods promoted autophagy and reduced apoptosis [239,240]. Although individuals in the long-term treatment group experienced improved muscle regeneration, motor function loss and respiratory insufficiency were not changed [241]. These studies clearly implicate impaired autophagy and mitochondria-mediated myofiber apoptosis in the pathology of UCMD and other ColVI dystrophies but raise questions about the viability of CsA as a therapy.

## 7. Conclusions

Advances in genetic, genomic, and proteomic technologies have resulted in an acceleration in the unraveling of the molecular mechanisms underlying the histopathology of more than 50 types of muscular dystrophy and opened the door to an exploration of alternative therapeutic approaches to glucocorticoid suppression of fibrosis, fat accumulation, and chronic inflammation associated with most muscular dystrophies. Strategies with broad promise are those that reduce proinflammatory NF-κB and TNF-α signaling. Studies in mouse models have shown success in ameliorating the pathology upstream of their expression through antioxidants (e.g., COX enzyme inhibitor) [44], suppression of the inflammasome (e.g., P2X7 antagonists) [149] or directly through the inhibition of NF-κB activity (e.g., vamorolone) [186]. Alternatively, inhibition of the Akt/mTOR signaling pathway by rapamycin suppresses the immune response to injury by reducing the proliferation of immune cells [253]. This has shown potential for ameliorating fibrosis in *Lmna-* and *Ftkn*-deficient mice [94,159,160,161]. Recent reports predict that direct suppression of fibrotic deposition in fibroblasts at the site of injury by disrupting uPA/uPAR signaling via Serp-1 represents a strategy downstream of chronic inflammation to reduce fibrosis [32,50,51]. 

Though we have seen a growth in pharmacotherapeutic strategies to slow down muscle loss through immunosuppression, the number of approved treatments for muscular dystrophy patients remains limited. The rarity of many of these diseases has served as an unfortunate barrier to the clinical trials essential for FDA approval [128]. To properly control the trial, characteristics like a definitive genetic diagnosis, age, sex, and stage of muscle wasting are required for inclusion. This requires establishing large multicenter trials over an extended period to recruit enough participants to be able to generate statistically significant results. Innovative off-label use of existing FDA-approved drugs, such as vamorolone and rapamycin, may reduce some of the challenges. Vamorolone is an example of a drug currently in clinical trials for the more prevalent DMD. 

## Figures and Tables

**Figure 1 biomolecules-13-01536-f001:**
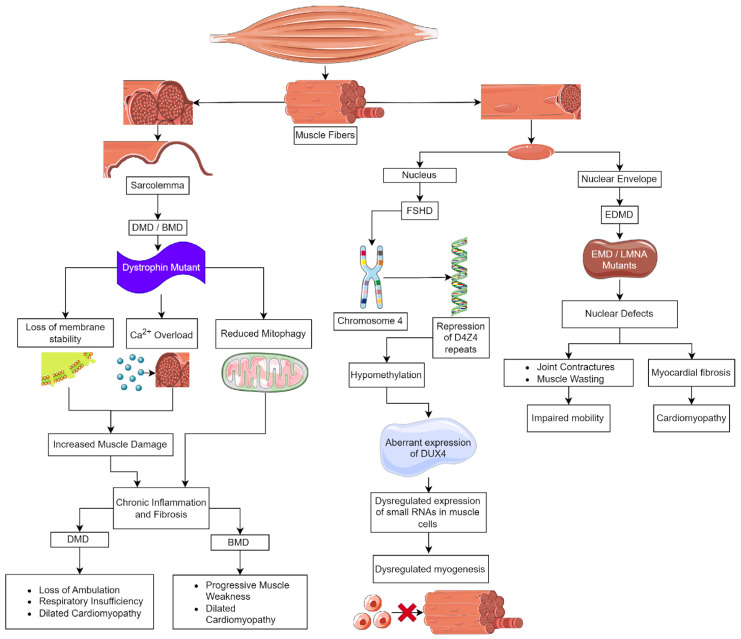
Schematic of the etiology of DMD/BMD, FSHD, and EDMD.

**Figure 2 biomolecules-13-01536-f002:**
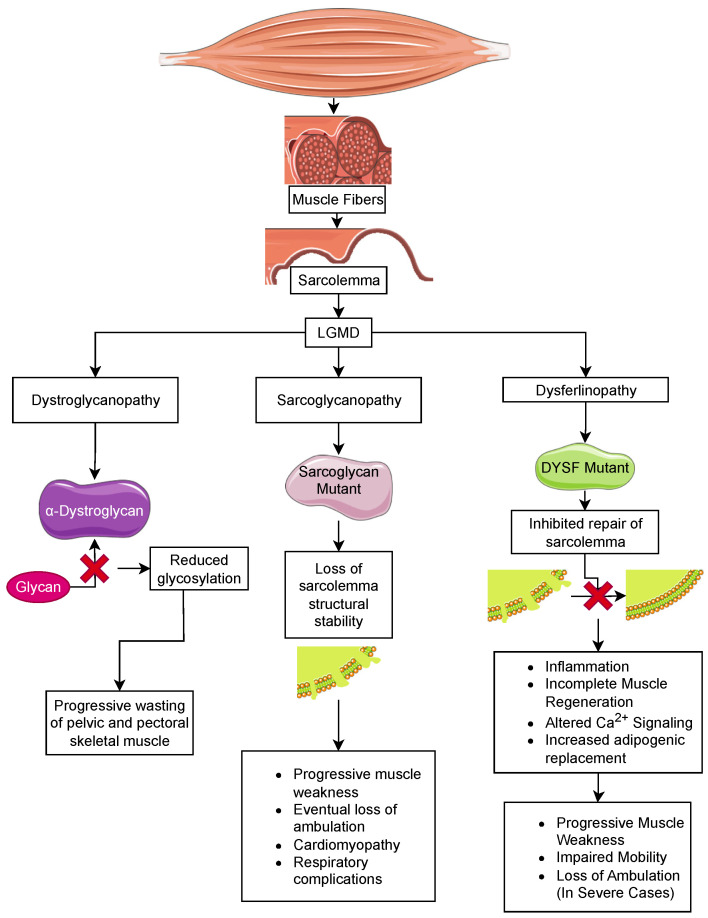
Schematic of the etiology of subgroups of LGMDs.

**Figure 3 biomolecules-13-01536-f003:**
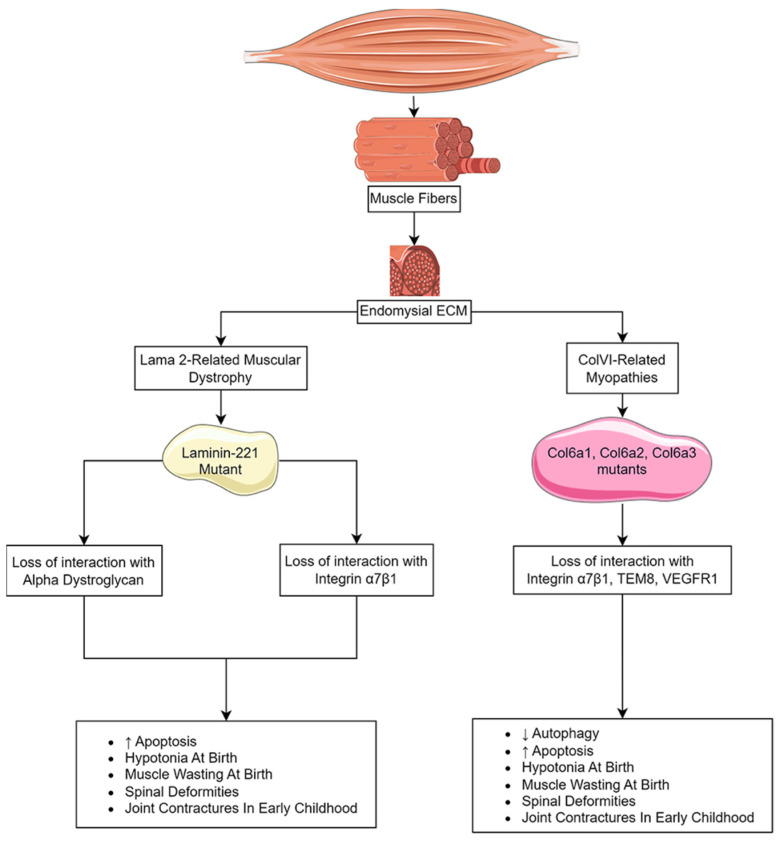
Schematic of the etiology of congenital muscular dystrophy associated with Laminin-221 and collagen VI mutations.

**Table 2 biomolecules-13-01536-t002:** Treatment strategies for EDMD and FSHD.

Treatment	Strategies	Outcomes	^1^C/PC	Refs
Emery–Dreifuss Muscular Dystrophy				
Theophylline	HDAC3 inhibitor	Rescued myotube formation in *EMD^−/−^* mice	PC	[86]
PD098059,	Inhibit MEK1 activation of MAPK	↓ ventricular dilation ↑ cardiac function in *Lmna^H222P/H222P^* mice	PC	[87]
Selumetinib	Inhibit MEK1 activation of MAPK	↓ ventricular dilation ↑cardiac function in *Lmna^H222P/H222P^* mice	PC	[88]
ARRY-371797	p38MAPK inhibitor	Phase II trial *LMNA* cardiomyopathy	C	[88]
Temsirolimus	mTORC1 inhibitor	↑ autophagy ↓ ventricular dilation no effect cardiac fibrosis	PC	[88]
Paclitaxel	Stabilize conductance in cardiomyocytes	↑ Cx43 in intercalated discs *Lmna^H222P/H222P^* mice	PC	[89]
Facioscapulohumeral Muscular Dystrophy				
Albuteral	β2-adrenergic agonist, inhibits DUX4 expression	Clinical trials -modest results	C	[90,91]
Losmapimod	MAPK inhibitor, blocks DUX4 transcription	Double-blind phase IIb trial ↑ anti-inflammatory ↓ fatty replacement	C	[92]

^1^C—clinical; PC—preclinical.

**Table 4 biomolecules-13-01536-t004:** Treatment strategies for laminin-211 and collagen VI related dystrophies.

Treatment	Strategies	Outcomes	^1^C/PC	Refs
Lama2-Related Muscular Dystrophy				
Losartan	AT_1_R inhibitor	↓ fibrosis ↓ TGF-β activity in *dy^W^/dy^W^* and *dy^2J^/dy^2J^* mice	PC	[225,226,228]
Vemurafenib	mTORC/p70S6K inhibitor	↓ fibrosis ↓ TGF-β activity in *dy^3K^/dy^3K^* mice	PC	[229]
Holofuginone	Stat3 inhibitor	↓ fibrosis ↓ TGF-β activity in *dy^2J^/dy^2J^* mice	PC	[230]
Omigapil	Inhibitor of apoptosis through GAPDH-Siah	↓ fibrosis ↑ grip strength ↑respiration rate in *dy^2J^/dy^2J^* mice pharmacokinetics clinical trials completed	PC/C	[231,232]
MG-132	Proteosome inhibitor	↑ muscle strength, ↑ mobility, ↑ lifespan, ↓ fibrosis in *dy^3K^/dy^3K^* mice	PC	[233]
Bortezomib	Proteosome inhibitor	↑ weight, ↑lifespan, ↓ fibrosis in *dy^3K^/dy^3K^* mice	PC	[234]
3-MA	Autophagy inhibition	↑ muscle repair, ↓ autophagy markers ↓ apoptosis, ↓ fibrosis in *dy^3K^/dy^3K^* mice	PC	[235]
NAC	Antioxidant	↑ muscle protection in * dy^2J^/dy^2J^ * mice	PC	[236]
Vitamin E	Antioxidant	↑ muscle protection in *dy^2J^/dy^2J^* mice	PC	[237]
Collage VI-Related Dystrophies				
Cyclosporin A	Inhibit MPTP in mitochondrial membranes	↓ apoptosis, ↑ autophagy in *Col6a1^−/−^* mice ↑ muscle regeneration, no change in loss of motor function, and respiratory insufficiency	PC/C	[238,239,240,241]

^1^C—clinical; PC—preclinical.

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
