# Peer review of "Pharmacotherapeutic Approaches to Treatment of Muscular Dystrophies"

_biomolecules, 2023, doi:10.3390/biom13101536_

Round 1
Reviewer 1 Report
This is a comprehensive review of therapeutic approaches that are currently being investigated and tested in clinical trials as alternatives to the commonly used glucocorticosteroid treatment to act on the chronic inflammation and fibrosis that are present in muscular dystrophies and contribute to the pathomechanisms of the disease. The article provides an update of the many data available in this field, with a review of approaches being considered, clinical trials carried out in humans, and progress or failures recorded. This is a very well-written article, with carefully designed and informative illustrations.
I have only a few minor comments:
1- Page 2, line 50: “There is a loss of ambulation in the teens to early twenties”. DMD patients are usually wheelchair bound before their twenties, most often before the age of 13.
2- Page 2, line 60: “The DMD has 97 exons” is not correct. The DMD gene contains 79 exons and 6 additional Exon 1 that determine the expression of long and short tissue-specific dystrophin isoforms.
Page 2, line 78-80: the paragraph describing the classes of DMD mutations may lack some clarity. “In DMD, many mutations are frameshifts resulting in premature termination or dysfunctional proteins. However, there are also in-frame deletions, duplications, and large deletions as well”. What is the difference in in-frame deletions and large deletions?
In the DMD gene, around 70-80% are actually large, multi-kilobase rearrangements (CNVs) consisting of large deletions (most frequent) or large duplications of one or more exons. The remaining 20-30% are small mutations (SNVs) that modify only one or a few nucleotides. As the authors point out, the impact of mutations on the reading frame determines the severity of the resulting phenotype: DMD caused by mutations (CNVs or SNVs) that introduce a premature stop codon, or the milder BMD, when mutations maintain an open reading frame and the expression of a residual amount of dystrophin.
Figure 1: it would be appropriate to add "Dilated cardiomyopathy" to "Progressive muscle weakness" for BMD, as dilated cardiomyopathy is the leading cause of death in BMD patients.
Page 9, line 333: change “following myotonic” to “myotonic dystrophy type 1” and a reference.
Page 14, line 518: what does the term "polymorphic mutations" mean? The term polymorphism refers to DNA variants present in the general population at relatively high frequencies and generally without pathological consequences.
Author Response
We would like to thank Reviewer 1 for their thoughtful comments and suggested edits. We have made changes in response to these comments that we feel strengthen the manuscript. The specific changes are outlined below.
1- Page 2, line 50: “There is a loss of ambulation in the teens to early twenties”. DMD patients are usually wheelchair bound before their twenties, most often before the age of 13.
This now reads: “There is a loss of ambulation in childhood to early teens and death in early adulthood.”
2- Page 2, line 60: “The DMD has 97 exons” is not correct. The DMD gene contains 79 exons and 6 additional Exon 1 that determine the expression of long and short tissue-specific dystrophin isoforms.
The sentence now reads “DMD is the largest gene in the human genome, it has 79 exons, the full-length transcripts include unique tissue specific first exons…” .
3-Page 2, line 78-80: the paragraph describing the classes of DMD mutations may lack some clarity. “In DMD, many mutations are frameshifts resulting in premature termination or dysfunctional proteins. However, there are also in-frame deletions, duplications, and large deletions as well”. In the DMD gene, around 70-80% are actually large, multi-kilobase rearrangements (CNVs) consisting of large deletions (most frequent) or large duplications of one or more exons. The remaining 20-30% are small mutations (SNVs) that modify only one or a few nucleotides. As the authors point out, the impact of mutations on the reading frame determines the severity of the resulting phenotype: DMD caused by mutations (CNVs or SNVs) that introduce a premature stop codon, or the milder BMD, when mutations maintain an open reading frame and the expression of a residual amount of dystrophin.
This paragraph has been rewritten and we believe that the writing is clearer.
4- Figure 1: it would be appropriate to add "Dilated cardiomyopathy" to "Progressive muscle weakness" for BMD, as dilated cardiomyopathy is the leading cause of death in BMD patients.
This has been added to figure 1.
5- Page 9, line 333: change “following myotonic” to “myotonic dystrophy type 1” and a reference.
This has been done, thanks for catching the editing error.
6-Page 14, line 518: what does the term "polymorphic mutations" mean? The term polymorphism refers to DNA variants present in the general population at relatively high frequencies and generally without pathological consequences.
The term has been removed and the sentence now reads: “ There are more than 260 mutations in DYSF reported to result in three types of muscular dystrophies…” .
Reviewer 2 Report
Rawls and colleagues provided a well-written and illustrated review. The authors systematized the known data on the mechanisms of occurrence and development of the most common types of muscular dystrophies, and also showed approaches to their therapy. However, I recommend the authors to supplement the text with important information:
1. Dystrophin also coordinates the assembly of ion channels and signaling molecules, and its loss leads to dysregulation of muscle cell ion channels and disruption of calcium, sodium, and potassium ion homeostasis. This must be specified. Recent review papers describe this function of dystrophin well.
2. Lines 227-238. Other approaches that mitigate the development of mitochondrial dysfunction in DMD should also be indicated. This applies to the use of calcium-dependent mitochondrial pore inhibitors (alisporivir, CsA, NIM811, or isoxazoles), potassium channel activators that reduce oxidative stress, mitochondrial biogenesis activators (givinostat), mitochondrial respiration substrates. All these approaches have a positive effect on the course of dystrophin-deficient pathology.
3. The authors also need to describe the approaches used to correct the function of the sarcoplasmic reticulum, which plays an important role in the regulation of calcium homeostasis in muscles. Recently, a lot of information has accumulated on this topic.
Author Response
We would like to thank Reviewer 2 for their thoughtful response. We had not considered calcium overload and mitochondrial dysfunction as within the scope of the review. After evaluating the topics raised by the reviewer, we have added a new section that we feel significantly improves the manuscript. Specific changes are outlined below. . All changes are underlined in the manuscript.
- Dystrophin also coordinates the assembly of ion channels and signaling molecules, and its loss leads to dysregulation of muscle cell ion channels and disruption of calcium, sodium, and potassium ion homeostasis. This must be specified. Recent review papers describe this function of dystrophin well.
We have added a section focused on these ion channels and potential therapeutics that target them. As the reviewer notes, there are a considerable number of recent reviews that cover this topic, and we tried to not overlap too significantly.
- Lines 227-238. Other approaches that mitigate the development of mitochondrial dysfunction in DMD should also be indicated. This applies to the use of calcium-dependent mitochondrial pore inhibitors (alisporivir, CsA, NIM811, or isoxazoles), potassium channel activators that reduce oxidative stress, mitochondrial biogenesis activators (givinostat), mitochondrial respiration substrates. All these approaches have a positive effect on the course of dystrophin-deficient pathology
We have added mitochondrial specific discussions to two areas of the manuscript. We have included alisporivir, which is a CsA analog. However, as the focus of this review is therapeutics that play a role in reducing inflammation and fibrosis, we did not include givinostat. In fact, part of the clinical trial inclusion criteria for this drug was a requirement that the patients be taking prednisone. NIM811 is a specific treatment for collagen VI mutations that cause MD, which is a rare and complex disease and for these reasons we did not include this particular type of dystrophy in this review.
- The authors also need to describe the approaches used to correct the function of the sarcoplasmic reticulum, which plays an important role in the regulation of calcium homeostasis in muscles. Recently, a lot of information has accumulated on this topic.
We have included some discussion of sarcoplasmic reticulum. Again we tried to focus on therapeutics that have a clear function in reducing or preventing inflammation or fibrosis.
Round 2
Reviewer 2 Report
The authors responded adequately to my comments. I also agree with their objections.
A couple of things that need to be corrected before publishing:
1. Table 1. Should be Alisporivir instead of Alisporrivir.
2. Lines 174-176. Authors should provide references to original works regarding the role of SERCA and the MCU. The reviews cited by the authors only summarize the known data. This is not exactly good practice.
